

# Shipping and algae emissions have a major impact on ambient air mixing ratios of NMHCs and methanethiol on Utö island in the Baltic Sea

Heidi Hellén[1], Rostislav Kouznetsov[1], Kaisa Kraft[2], Jukka Seppälä[2], Mika Vestenius[1], Jukka-Pekka Jalkanen[1], Lauri Laakso[1,3], Hannele Hakola[1]

[1] Finnish Meteorological Institute, P.O. Box 503, FI-00101 Helsinki, Finland
[2] Finnish Environment Institute, Latokartanonkaari 11, FIN-00790 Helsinki, Finland
[3] Atmospheric Chemistry Research Group, Chemical Resource Beneficiation, North-West University, Potchefstroom 2520, South Africa

*Correspondence to*: Heidi Hellén (heidi.hellen@fmi.fi)

**Abstract.** Mixing ratios of highly volatile organic compounds were studied on Utö Island in the Baltic Sea. Measurements of non-methane hydrocarbons (NMHCs) and methanethiol were conducted using an in situ thermal desorption-gas chromatograph-flame ionization detector/mass spectrometer (TD-GC-FID/MS) from March 2018 until March 2019. The mean mixing ratios of NMHCs (alkanes, alkenes, alkynes, aromatic hydrocarbons) were at the typical levels for rural/remote sites in Europe and as expected the highest mixing ratios were measured in winter while in summertime, the mixing ratios remained close to or below detection limits for most of the studied compounds. Sources of NMHCs during wintertime were studied using positive matrix factorization (PMF) together with wind direction analyses and source area estimates. Shipping was found to be a major local anthropogenic source of NMHCs with a 21% contribution. It contributed especially on ethene, propene and ethyne mixing ratios. Other identified sources were gasoline fuel (15%), traffic exhaust (14%), local solvents (6%), and long-range transported background (42%). Contrary to NMHCs, high mixing ratios of methanethiol were detected in summertime (July mean 1000 pptv). The mixing ratios followed the variations of seawater temperatures and sea level height and were highest during the daytime. Biogenic phytoplankton or macroalgae emissions were expected to be the main source for methanethiol.

## 1 Introduction

Atmospheric NMHCs (non-methane hydrocarbons) are composed of light alkanes, alkenes, alkynes and aromatics with high vapor pressure. Together with compounds containing additional oxygen, nitrogen or other heteroatoms, they are also called volatile organic compounds (VOCs). NMHCs are emitted to the atmosphere due to fossil fuel or wood burning, solvent usage,



gas leakage etc. In the atmosphere NMHCs react with nitrogen oxides by means of hydroxyl radical reactions leading to the production of ozone. Ozone is a phytotoxic compound that is also harmful to health. Therefore, the EU has set limit values for
ozone concentrations, and implements policies to reduce emissions of ozone precursors. During 1994-2020 the emissions of NMHC in the European Union decreased by ~58% (EMEP emission database, https://www.ceip.at/webdab-emission-database, last accessed 9.10.2023).

NMHCs were measured on Utö Island in the Baltic Sea in Finland during 1992-2007 by collecting air samples in stainless steel canisters twice a week for further analysis in the laboratory. The results of these measurements have been published in
Laurila and Hakola (1996) and Hakola et al. (2006). These studies found a clear seasonal cycle of the NMHCs with the highest mixing ratios during winter. In this study, from March 2018 to March 2019, NMHCs were measured in the same location but at higher frequency, with a 2-hour time resolution and using in situ gas-chromatograph. With higher time resolution data, it is possible to study source the apportionment and source areas of measured NMHCs.

Positive matrix factorization (PMF) is commonly used for source studies of air pollutants (Sun et al. 2020). It has also been
applied for NMHCs at different kinds of locations but only a few studies have used it on NMHCs at remote/rural sites (Leuchner et al. 2015, Lanz et al. 2009, Sauvage et al. 2009). Vestenius et al. (2021) applied PMF for more reactive biogenic VOCs in a boreal forest. At these kinds of environments photochemical aging must be considered when interpreting the results (Yuan et al. 2012). In this study, we use PMF combined with wind direction distributions and source area analyses to study sources of NMHCs in marine air on Utö Island in the Baltic Sea. In Northern Europe, the source areas of NHMCs have been
studied earlier at a sub-Arctic site, i.e., Pallas, Finland (Hellén et al., 2015), where source area studies of NMHCs indicated that the EU was no longer a significant source area for NMHCs at that study site. The main source area was in Eastern Europe to the southeast of Pallas.

Together with other pollutants (e.g., particles, sulphur dioxide and nitrogen oxides) combustion in ship engines produces NMHCs and these emissions may have strong impacts on the air quality in coastal and marine areas (e.g., Viana et al. 2014,
Tang et al. 2020). One aim of this study was to quantify the impact of shipping on mixing ratios of NMHCs in marine air in the Baltic Sea.

In the current publication we also study how frequently occurring phytoplankton blooms may affect the atmosphere. The Baltic Sea is a brackish, eutrophied, and non-tidal coastal sea with high concentrations of dissolved organic matter (DOM), where phytoplankton blooms occur frequently (Kahru and Elmgren, 2014). Kilgour et al., (2021) studied sulphur emissions during
an induced phytoplankton bloom and they found that dimethylsulfide (DMS), methanethiol ($CH_3SH$), and benzothiazole ($C_7H_5N_S$) account for on average over 90% of total gas-phase sulphur emissions. While there are lots of studies on the mixing ratios of DMS, less is known about the other sulphuric compounds in marine air. Lawson et al. (2020) measured methanethiol and DMS in the air over remote parts of the southwest Pacific Ocean and Novanak et al. (2022) their fluxes and mixing ratios in marine air on the coast of California, US (Novanak et al. 2022). Sulphur compounds are oxidized in the air forming sulphur
dioxide ($SO_2$). In the air, $SO_2$ is further oxidized producing sulphuric acid, which participates in new particle and cloud formation in the air.



## 2 Experimental setups

### 2.1 Measurement site

The Utö Atmospheric and Marine Research Station of the Finnish Meteorological Institute (59º 46'50N, 21º 22'23E) is located at the outermost edge of the Archipelago Sea, facing the Baltic Sea proper. The station provides an excellent possibility to observe marine biogenic emissions with minimal interference from terrestrial sources. The station produces real-time, high-frequency measurements of the physical, chemical and biological features of the water column and atmospheric concentrations of trace gases and aerosols (Fig. 1a; Laakso et al., 2018; Kraft et al., 2021; Honkanen et al., 2018, 2021, 2023; Rautiainen et al, 2023).







**Figure 1: a) Description of the Utö marine research station and b) predicted CO₂ emissions from ships sailing the Baltic Sea during 2022 (map © 2023 Google - Image Landsat / Copernicus). The main shipping routes are coloured according to emissions in relative mass units per unit area.**

## 2.2 Measurements of NMHCs and methanethiol


The samples were collected from a 9 m long mast (inlet 12 m above sea level, Fig. 1, Marine research station 1) with flow rate of ~2.2 L min⁻¹. The mast is located approximately 5 meters from the sea on the western edge of Utö island. A subsample was collected to a gas-chromatograph through a heated line. An in-situ thermal desorption unit (Unity 2 + Air Server 2, Markes International ltd.) connected to a gas chromatograph (Agilent 7890) with a mass spectrometer (Agilent 5975C) and a flame

ionization detector (TD-GC-MS/FID) was used. Two columns (HP-1, 50 m*0.22 mm*1 µm and Al/Na₂SO₄ PLOT, 50 m*0.32 mm) were connected using an Agilent Deans switch. Samples were taken every other hour from a 35 m long fluorinated ethylene propylene (FEP) inlet (1/8 inch I.D.) located on the 9 m long mast. An extra flow of 2.2 L min⁻¹ was used to avoid losses of the compounds on the walls of the inlet tube. Samples were collected directly from this ambient air flow through a Nafion drier into the cold trap (U-T17O3P-2S, Markes International Ltd.) of the thermal desorption unit. The sampling time

was 30 min and the sampling flow through the cold trap 20 ml min⁻¹. All the lines and valves in the thermal desorption unit were kept at 200ºC. During sampling, the cold trap was kept at -30ºC. For desorption, the cold trap was heated to 300ºC for 3 minutes and flushed with a helium flow of 10 ml min⁻¹. Every 50th sample was a calibration sample. The calibration gas (National Physical Laboratory, 30 VOC mix) contained C₂-C₈ alkanes, C₂-C₅ alkenes, C₆-C₉ aromatic hydrocarbons, ethyne and isoprene at a ~4 ppb level.

In spring and summer we detected an unknown compound on the flame-ionization detector. It was later identified as methanethiol. Methanethiol was not included in the used standard mixture, but it was later identified based on an authentic standard (Linde gas/BOC, 5 component mix at a 10 ppm level) and it was also detected from cyanobacteria culture samples grown in the laboratory. It was calibrated as close eluting pentane and corrected based on carbon number.

## 115 2.3 Positive matrix factorization (PMF) modelling

The Utö winter NMHC dataset was analysed using EPA PMF version 5.0. PMF (Positive matrix factorization) which is a receptor modelling tool that is widely used in source apportionment of air pollution (Hopke, 2016; Hopke et al., 2020). PMF decomposes the measured time series data matrix into two matrices, i.e., factor contributions and factor profiles and then uses

a chemical mass balance equation to find a number of user specified factors (potential sources) that affect the measured species' concentration at the receptor. The results are constrained to be non-negativity for the factor profiles and non-significant negativity for the factor contributions. VOC concentrations and especially biogenic VOC emission profiles may change quite rapidly in the atmosphere on the way from emissions to the receptor site due to the high reactivity of VOCs. Prior works on



source apportionment of biogenic VOCs have shown that PMF is a valuable tool also for this kind of data (e.g., Vestenius et al., 2021). Even though studied anthropogenic NMHCs have longer atmospheric lifetimes than those of biogenic VOCs, their ratios may still vary during the transport, and this has to be taken into account while interpreting the results.

Uncertainty calculations for the PMF model were made according to Polissar et al. (1998). The expanded measurement uncertainty (U) was estimated from partial uncertainties of analytical precision, standard preparation, and sampling flow. As PMF does not tolerate missing values in the data matrix, missing values (e.g., missing species in the sample) were replaced by species median with their uncertainties multiplied by ten, so that this "synthetic" data would not have effect on the model resolution. Values below detection limit (<LOD) were replaced by 0.5*LOD and their respective uncertainties were set to 5/6*LOD. Species were categorized as "strong", "weak" or "bad" using their sample to noise ratios. S/N species with ratios over 2 were generally categorized as "Strong", and species with S/N values between 0.2 and 2 were generally categorized as "weak." Species with S/N values less than 0.2 were categorized as "bad" and were removed from the model. Weak categorization increases the species' uncertainty by the factor of three. However, only three of the 14 modelled species; ethane, ethyl benzene and toluene were categorized as "Strong", the other 11 species were categorized as "weak" due to their low S/N ratio. Uncertainty analyses of the model results were made using bootstrapping. The factor contributions (time series) were combined with local wind data using the OpenAir software package in R (Carslaw, 2018). A conditional bivariate probability function (CBPF) was used to estimate the conditional probabilities of source directions and distance from the receptor. CBPF takes wind speed into account as a third variable in addition to concentration and wind direction and gives a probability of direction and likely distance of high factor contributions (potential source). In CBPF, the 75th percentile was used as threshold value so that the analysis gives a probability of direction and likely distance of highest factor contributions.

## 2.4 Source area estimates

Adjoint atmospheric dispersion simulations were used to identify the source areas for the samples collected during the measurements. The method is quite similar to the one used by Meinander et al. (2013), Hellén et.al. (2015) and Meinander et al. (2020), but for the sake of completeness, we briefly describe it below.

The simulations were performed with SILAM v5_8 (System for Integrated modeLing of Atmospheric coMposition http://silam.fmi.fi, accessed on 21.8.2023). This system incorporates a Eulerian non-diffusive transport scheme (Sofiev et.al., 2015). The model has been extensively validated for a variety of atmospheric dispersion and source inversion problems on a regional and global scale (Petersen et. al., 2019; Kouznetsov et.al. 2020)

The model was driven with the meteorological fields from ERA-5 (Hersbach et. al. 2020) at the resolution of 0.25x0.25 degrees. Adjoint simulations were made on a 0.5°x0.5° resolution grid, 8 vertical layers with varying thicknesses, spanning from 30 m at the surface to 2000 m at altitudes up to 6 km. For each of the about 900 samples collected, the model was integrated for six days backwards in time on a domain covering northern Europe. The resulting fields of sensitivity distribution (also known as footprints or retroplumes) were aggregated in time and stored for further analysis.



Each measured sample combines contributions from spatially and temporally distributed sources. Adjoint simulations allow for reconstructing a 4D sensitivity pattern of each sample to the location of a source in space and time. Combining the
sensitivities with the sampled values one can infer the likely source areas for a specific variable. Since the individual observed compounds are highly correlated in the samples in this study, we used the concentrations of the PMF factors, rather than concentrations of individual compounds. For each of the factors, the retroplumes were aggregated for below the 20th percentile and above the 80th percentile to get a typical "clean"-sample source area and "polluted"-sample source areas. The aggregated retroplumes show the relative sensitivity of the corresponding samples to various locations of the emission source. The areas
that are high for polluted samples, but low for clean ones are likely the origin of the specific factor.

## 2.5 Complementary data

Total phytoplankton biomass was obtained using imaging flow cytometry. An Imaging FlowCytobot (IFCB, McLane Research
Laboratories, Inc., United States) was connected to the Utö station's flow through system, taking a 5-mL sample approximately every 20 minutes. A chlorophyl *a* trigger was used to target the phytoplankton community (detailed explanation of the sampling system in Kraft et al. 2021). The data was classified using a Convolutional Neural Network classifier and image specific biovolumes were computed (Moberg and Sosik 2012, Kraft et al. 2022). The total phytoplankton biomass was calculated by summing up the total biovolumes of all classes, including also unclassified images, and converting the total biovolume ($\mu m^3$
$mL^{-1}$) to total biomass ($\mu g\ L^{-1}$) by assuming a plasma density of 1 g cm-3 (CEN, 2015).
The used meteorological data and ozone ($O_3$), sulphur dioxide ($SO_2$), particulate matter <2.5 µm ($PM_{2.5}$) and nitrogen dioxide ($NO_2$) concentration data is available at the Finnish Meteorological Institute open access data portal (https://en.ilmatieteenlaitos.fi/download-observations, last accessed 21.6.2023). The data were collected at a close by atmospheric research station on Utö island. Sea surface temperature was measured at a wave buoy 60 km to the South-West
of Utö with a Datawell DWR4 Waverider buoy. Sea level height at Utö was interpolated from tide gauge observations at Föglö (62 km from Utö) and Hanko (90 km from Utö). The values are given relative to the theoretical mean sea level.
Carbon dioxide ($CO_2$) emissions originating from ship traffic over the Baltic Sea were used for presenting main shipping routes (Fig 1b). The emissions were modelled using the Ship Traffic Emission Assessment Model (STEAM), which uses Automatic Identification System data to describe ship traffic activity. The method is presented elsewhere (Jalkanen, 2009; Jalkanen et al.,
2012; Johansson et al., 2017).

## 3 Results and discussion
### 3.1. Seasonal variations of NMHCs

The seasonal variations of NMHCs followed a well-known cycle with maximum mixing ratios during the dark wintertime and minimum mixing ratios during summer, when mixing ratios of many compounds were below detection limits due to effective





sink reactions with OH radicals (Fig. 2). Generally, the more reactive the compounds are, the larger the amplitude between summer and winter mixing ratios is. The mean mixing ratios were at typical levels for rural/remote sites in Europe (Solberg et al. 2020).








**Figure 2: Monthly mean box and whisker plots of measured mixing ratios (pptv). The boxes represent second and third quartiles and the vertical lines in the boxes median values. The whiskers show the highest and the lowest measurements.**



**3.2 Source apportionment of NMHCs**


In PMF modelling, different combinations of 4 to 7 source factor solutions were tested. The most reasonable and interpretable results were obtained by a 5-factor solution, which was chosen for further investigation. Also, bootstrapping showed acceptable results to all factors for this solution, with no swaps for factors 1 to 4 and 95% mapping for F5, which is well within the acceptable range (80%). One of the factors represents regional background air and the other four factors were interpreted as

ship, local solvent, gasoline, and traffic exhaust emissions as represented in Table 1 and described below. For PMF analysis, we could only use winter data since the mixing ratios of many of the compounds were very low or below detection limits during other times. The profiles of factors are not expected to directly represent the profiles of real emissions, since most of these emissions are transported to the site over long distances, and during transportation NMHCs are oxidized with different rates, which means that the ratios of the compounds may change. This has been considered when interpreting the results.

Ethane has a relatively high contribution to all factors (Fig. 3a). Ethane is the longest living NMHC and has the highest mixing ratios. Background factor F3 is a major contribution factor for ethane (Fig. 3b). Due to the high mixing ratios of ethane compared to other compounds (Fig. 2), small changes in ethane mixing ratios (even within uncertainties) may result in high contributions to the other factors. This has also been considered when interpreting the results.

**Table 1: Identification of the PMF factors and the mean contribution to the NMHC mixing ratios measured at the site.**

| Factor | Name | Driver | Contribution |
| --- | --- | --- | --- |
| F1 | Gasoline fuel | Butane, pentanes | 15% |
| F2 | Traffic exhaust | Aromatic hydrocarbons | 14% |
| F3 | Background | Ethane, propane, stable contribution | 42% |
| F4 | Ship emissions | Ethene, propene, ethyne | 21% |
| F5 | Local solvent | Ethylbenzene, o-xylene | 6% |

**3.2.1 Identification of the PMF factors**


Factor 1 (F1) was identified as gasoline fuel emissions. The factor had high contribution to butanes and pentanes (Fig. 3). These compounds are known to make a major contribution on the NMHC emissions of gasoline fuel and especially evaporated gasoline (Hellén et al. 2006). Wind direction distribution indicated that the main source for this factor was further away to the east of the measuring site (Fig. 4). This direction includes the main harbours in the Gulf of Finland and the city of Saint

Petersburg. This is supported by the sensitivity maps (Fig. 5). The maps corresponding to the clean samples for F1 (gasoline)



have clear gaps over Russia and the southern shore of the Gulf of Finland, which correspond to high values in the polluted maps (Fig. 5ab). These areas likely contribute to high concentrations of this factor.

Factor 2 (F2) was a traffic exhaust emission factor. It was characterized by the strong contribution of aromatic hydrocarbons (Fig. 3), which are known to be major compounds in traffic exhaust emissions (e.g., Hellén et al. 2006, Wang et al. 2020, Wu et al. 2020a). This factor made a high contribution especially in the beginning of February during high wind speeds (Fig. 6). Wind direction distribution indicated that the main source was to the east of Utö in the direction of Saint Petersburg (Fig. 4). Sensitivity maps of F2 showed very similar source areas as for F1 (gasoline) with contribution from Russia and the southern shore of the Gulf of Finland, except that it also contains a substantial contribution from the Baltic states (Fig. 5).

Factor 3 (F3) was identified as long-range transported background air with a high contribution of the longest living NMHCs, ethane and propane (Fig. 3). The contribution of the factor over time was relatively stable compared to other factors (Fig. 6) and the highest contribution of the measured NMHCs, 42%, came within this factor. Wind direction distribution was more scattered as expected for the regional background (Fig. 4), but the main source area was to the west and north-west from the site. The close by shipping routes going to the Gulf of Bothnia and between cities of Turku (Finland) and Stockholm (Sweden) are in that direction and ships running on those routes use liquified natural gas (LNG), which is known to have ethane and propane emissions (Anderson et al. 2015) and could be influencing this factor as well. Also, sensitivity maps indicated that F3 got most of its contribution from Finland and Sweden, and much less contribution from the Baltic states and Russia (Fig. 5).

Factor 4 (F4) was identified as a shipping emission factor. It was characterized by the high contribution of ethene, propene and ethyne, which are major NMHCs in ship emissions (Bourtsoukidis et al., 2019; Wu et al., 2020b). The variation of the factor contribution followed the variation of $NO_2$ and $PM_{2.5}$ (Fig. 6). Based on the wind direction distribution the main source area coincided with the main ship route going to the Finnish and Russian harbours in the Gulf of Finland (Fig. 1b). Sensitivity maps of air masses (Fig. 5) also indicated a main contribution from southern and south-eastern sectors with quite uniform directional spawn. This ship emission factor made a 21% contribution to the total measured NMHCs.

Factor 5 was interpreted as a local solvent factor. It was a significant source only of ethylbenzene and o-xylene (Fig. 3b). These compounds are well-known solvents, e.g., in paints and coatings (e.g., Castano et al. 2019, Song and Chun 2021). However, F5 contribution to the total measured NMHCs was low, only 6%. The wind direction probabilities indicated the local origin of the source; to the direction of the village and harbour of Utö island during low wind (Fig. 4). Based on the sensitivity maps (Fig. 5) F5 had a similar pattern as F4 with a much narrower high-contribution sector in south-eastern direction from the measurement site.

The findings of the sensitivity maps were consistent with the wind analysis, being more specific for distant sources (F1-F3), for which the large-scale trajectories are important, and much less specific for local sources whose contribution is primarily controlled by local winds.



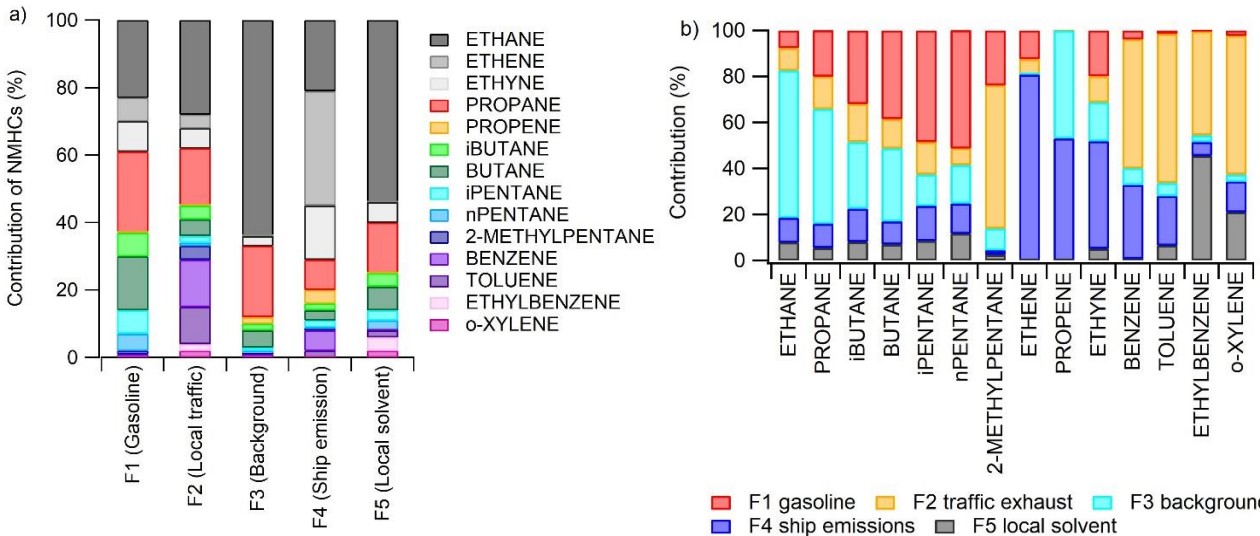


**Figure 3: a) Relative contributions of compounds' mixing ratios in factors (% of factor sum) and b) relative contribution of the factors on the average compound mixing ratios for the period of observations.**





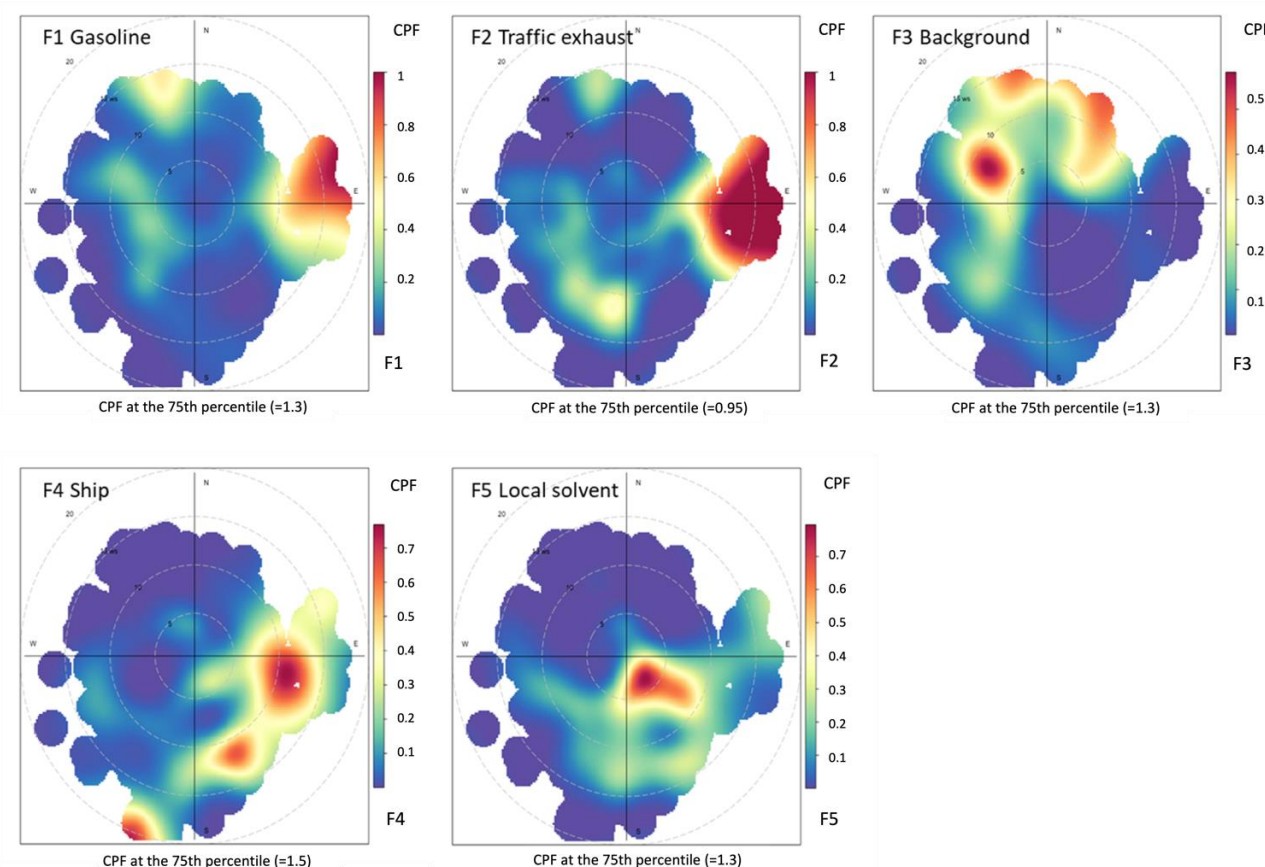


**Figure 4: Wind direction distributions corresponding with the factors.**



**Figure 5: Sensitivity maps for "low" and "high" factor loadings showing the source area probabilities of factors (F1-F5).**



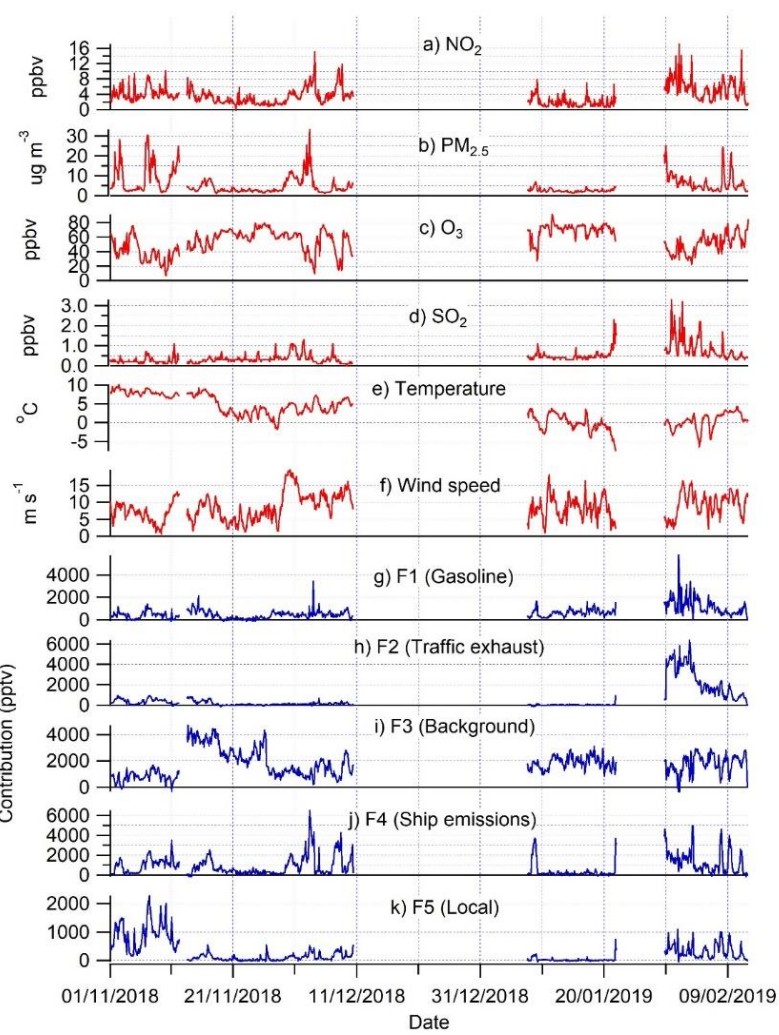

**Figure 6: Time series of air pollutants (a-d, in red), meteorological parameters (e-f, in red) and PMF factor contributions (g-k, in blue) at Utö during the NMHC measurement periods in winter 2018-2019.**


### 3.2.2 Sources of NMHCs

Based on the PMF analyses, the main source of studied NMHCs at Utö island was long-range transported air masses (F3) with a 42% contribution (Table 1). The strongest local/regional source was ship emissions F4 (21% contribution to the total

NMHCs) followed by gasoline fuel (F1) and traffic exhaust emissions (F2) with 15% and 14% contributions, respectively. In addition, there was a local solvent evaporation source (F5) with a 6% contribution.

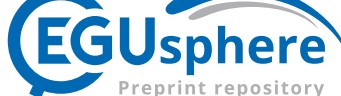

Source contributions of individual NMHCs were highly variable (Fig. 3b). For lightest alkanes, ethane and propane, long-range transported background air was clearly the main source with 64% and 50% contributions, respectively. This is expected since compounds with long atmospheric lifetimes are known to accumulate in the atmosphere in northern latitudes during

wintertime. Relatively high wintertime mixing ratios of them are detected even in remote areas (Hellén et al. 2015, Solberg et al. 2020). Gasoline fuel emissions (~35%) and background air (~30%) were major sources of butanes. For pentanes the main source was gasoline emissions with a ~50% contribution. For the alkane with shortest atmospheric lifetime, 2-methylpentane, traffic exhaust emissions were the main source with a 62% contribution.

Alkenes (ethene and propene) mainly originated from ship emissions with 81% and 53% contributions. For propene, long-

range transported background air was also significant source with a 47% contribution. Due to the relatively short atmospheric lifetime of propene this is not expected. Uncertainties on defining a proper blank value for propene may have induced this. For other compounds, a blank was not detected or was not as significant. Ethyne was only alkyne detected and ship emissions were the major source for it with a 47% contribution.

For aromatic hydrocarbons, traffic exhaust emissions were the major source with 45-65% contributions. Based on the wind

direction distribution, these emissions mainly originated from the east from the direction of the city of Saint Petersburg 500 km to the east from site. For benzene, toluene and o-xylene, ship emissions also resulted in 32%, 21% and 13% contributions, respectively. In addition, local solvent emissions contributed to the mixing ratios of ethylbenzene and o-xylene with 46% and 21% contributions, respectively. Due to shorter lifetime, these compounds have very low mixing ratios in remote areas and therefore as expected long-range transported background air did not result in a strong contribution.


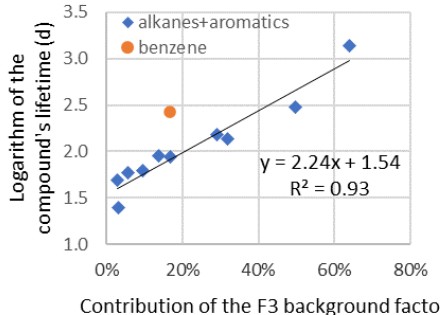

**Figure 7: Correlation of the logarithm of a compound's lifetime (Hellén et al. 2015) with the contribution of the F3 background factor to the mixing ratio of the compound.**


For alkanes and aromatic hydrocarbons, the loading of background factor F3 had a clear correlation ($R^2$=0.93) with the logarithm of the compound's lifetime (Fig.7). As expected, for compounds with the longest lifetime the contribution was highest. However, based on benzene lifetime, a stronger impact of the background air would have been expected. A higher





contribution of background air masses on benzene mixing ratios has been found even in the city of Helsinki in Finland (Hellén

et al. 2006). This could indicate a stronger local source. However, the mixing ratios of benzene were not high, and it had low

signal to noise ratio and therefore high uncertainties in the PMF solution and measurements of benzene were expected.

### 3.3 Mixing ratios of methanethiol

In addition to the NMHCs, we also detected methanethiol. Contrary to the usual annual cycle of anthropogenic VOCs, methanethiol had a maximum during spring and summer (Fig 2). Therefore, it is expected to have a biogenic origin. In earlier studies methanethiol has been detected, for example in phytoplankton and oceanic emissions (Kilgour et al. 2022, Novak et al. 2022). It is formed in the seawater from the same precursor metabolite, dimethyl sulfoniopropionate (DMSP), as DMS (Kiene and Linn, 2000).

At Utö, methanethiol mixing ratios started to increase in the end of April when the daily mean ambient air and seawater temperatures were above 5 and 4°C, respectively (Fig. 8a). The mixing ratios increased following changes in seawater temperature. The maximum mixing ratios were measured at the end of May. After mid-July, the mixing ratios started to decrease even though seawater temperature still increased (Fig. 8a). The mixing ratios declined below detection limits in September. Total phytoplankton biomass in seawater was measured concurrent with the atmospheric VOCs during July-August

and it is plotted together with methanethiol in Fig. 8b. Unfortunately, we do not have phytoplankton data from early summer due to instrument failure. There are many factors that affect the removal of substances from water to air and dilutions of emissions in the air, but the methanethiol mixing ratios seem to follow the total phytoplankton biomass in seawater and the decrease in phytoplankton biomass may explain the decrease in the mixing ratios after mid-July.

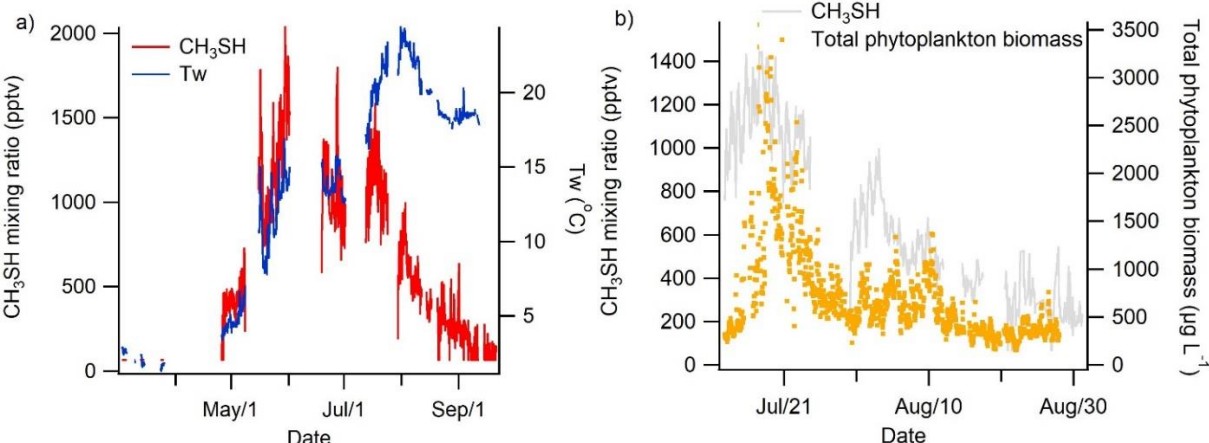


**Figure 8: Methanethiol (CH3SH) mixing ratios together with a) seawater temperature (Tw) in March – September and b) total phytoplankton biomass in July - August 2018 at Utö.**





The diurnal variation of the methanethiol mixing ratios followed the variation of the seawater temperature except in September,
       when mixing ratios were already close to the detection limit (Fig. 9). During the daytime, sink through OH oxidation is much
       higher than during the night. In addition, the mixing of layer heights in the atmosphere are expected to be clearly higher during
       the day resulting in higher dilution of emitted methanethiol during the daytime. Since the highest mixing ratios were still
       measured during the daytime, methanethiol is expected to have a very strong daytime emission source compared to nighttime
emission. Light induced emissions could explain this. However, no correlation with radiation was found.

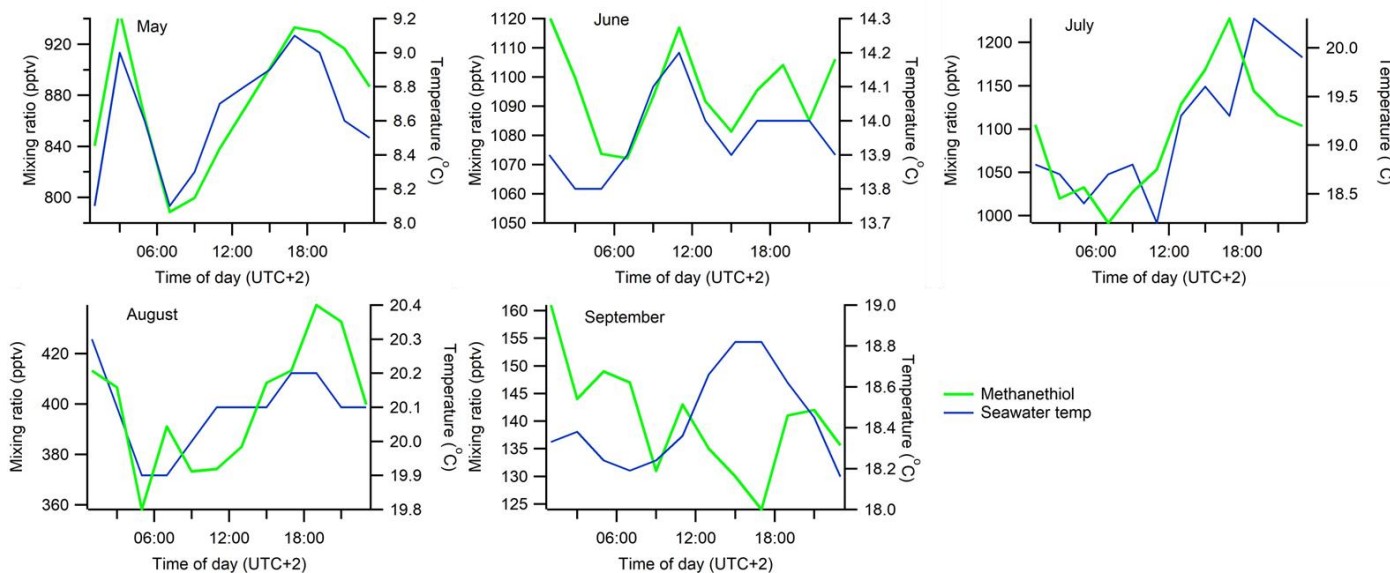

**Figure 9: Monthly mean diurnal variation of the methanethiol mixing ratios and seawater temperatures.**


       Biogenic emissions of volatile organic compounds from vegetation are generally known to have exponential dependence on
       air/leaf/needle temperature (Guenther et al. 2012). Here, dependence of the mixing ratios of methanethiol on both ambient air
       and seawater temperatures were studied (Fig. 10). The temperature dependence of the methanethiol mixing ratios varied over
       the season. In May, strong exponential dependence both on seawater ($R^2$=0.83) and ambient air ($R^2$=0.65) temperatures were
found. In June, the mixing ratios did not correlate with seawater or ambient air temperatures ($R^2$=0.02 and 0.03, respectively).
       However, the measurements were within quite a narrow temperature range. In July, there was some dependence with ambient
       air ($R^2$=0.4), but no correlation with seawater temperature ($R^2$=0.07). In August, the correlation of the mixing ratios with
       seawater and ambient air temperatures was again high with $R^2$ being 0.83 and 0.62, respectively. The reason for these
       differences could be shifts in phytoplankton community composition and their physiological status, possible contributions
from macroalgae vegetation from shoreline, and meteorology.




In addition to temperature, sea level height had a clear negative correlation with the methanethiol mixing ratios especially in May ($R^2$=0.69) and August ($R^2$=0.69, Fig. 10c). With lower sea levels, more macroalgae may be exposed to the ambient air and start decaying faster inducing more methanethiol emissions. This strong negative correlation could indicate macroalgae being a significant source of methanethiol. In June and July there was no correlation with sea level height, which indicates that

other sources (e.g., phytoplankton) played a more important role. While seawater and ambient air temperature dependencies (Fig 10 a and b) were different for May and August, sea level height followed the same curve during both months (Fig 10 c). The wind direction distribution (Fig. 11) indicates that the highest mixing ratios of methanethiol were measured during northernly winds (5 – 10 m s$^{-1}$). To the north of the site, there are wide archipelago areas with frequently occurring phytoplankton blooms and macroalgae as possible sources of methanethiol.


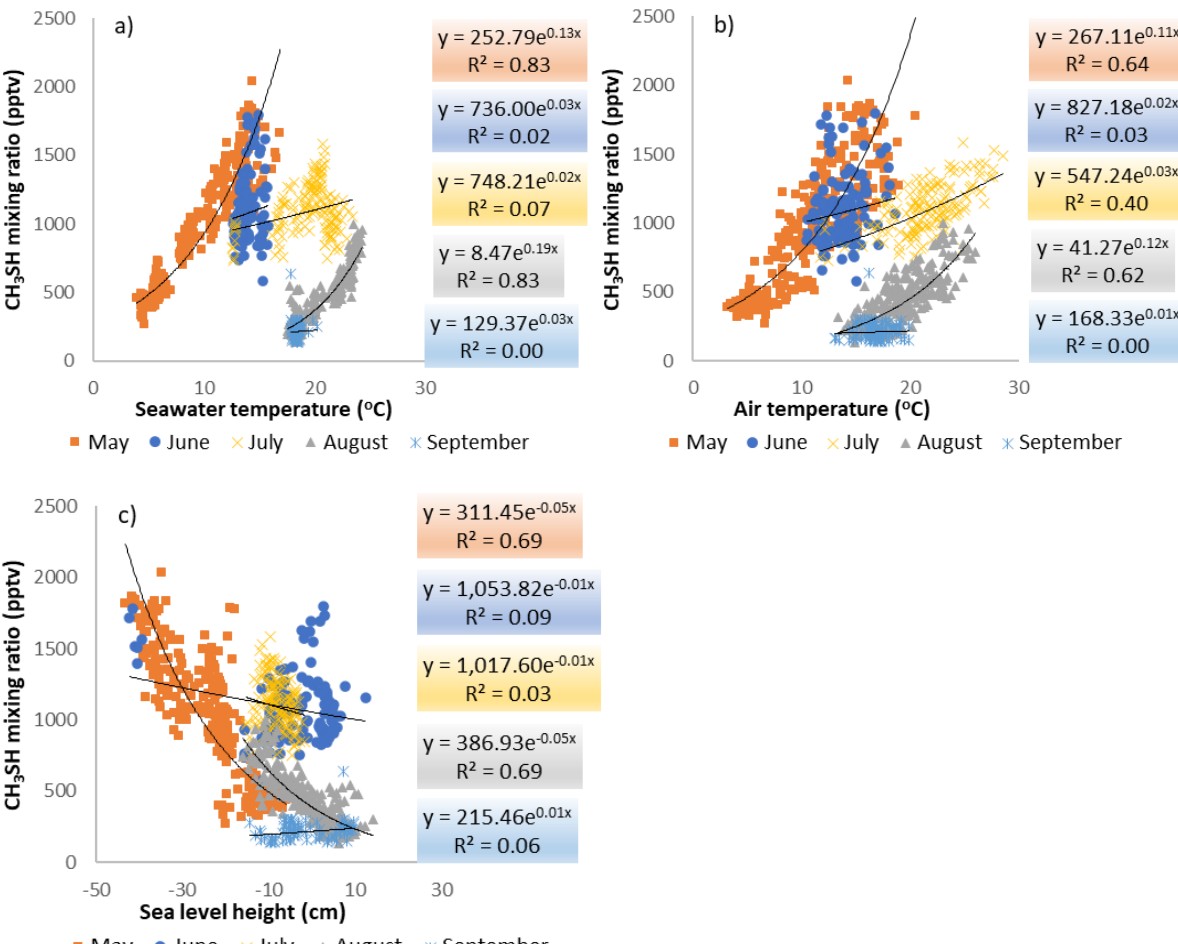

**Figure 10: Dependence of methanethiol (CH₃SH) mixing ratios on a) seawater temperature, b) ambient air temperature and c) sea level height.**




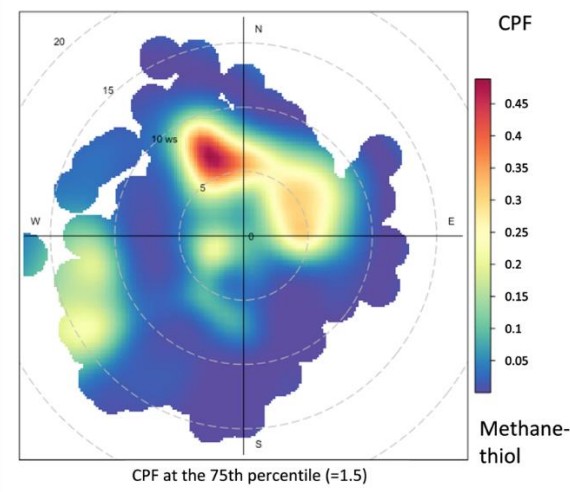

CPF at the 75th percentile (=1.5)

**Figure 11: Wind direction distribution for the measured methanethiol mixing ratios.**

There is very little information available on methanethiol mixing ratios or emissions globally. Novak et al. (2022) measured a
mean mixing ratio of 19 pptv at the Scripps Institution of Oceanography in La Jolla, CA, USA, during September 2019. The
highest measured value was 217 pptv. Lawson et al. (2020) measured a mean mixing ratio of 18 pptv in the remote
southwestern Pacific Ocean in the summer of 2012. These values are clearly lower than measured in this study in the Baltic
Sea where highest monthly means were ~1000 pptv. The coastal location with the adjacent macroalgae vegetation and high
production of phytoplankton in the nutrient rich and eutrophied Baltic Sea may explain these higher values.
Leck and Rodhe (1991) found 15 times higher DMS concentrations compared to methanethiol in seawater in the Baltic Sea.
Here we did not detect DMS, but our method was not optimized for it, and it is possible that we do not capture it with our
method. Methanethiol is intensively malodorous. It has an odour threshold of 1000-2000 pptv. During warm weather with
intensive phytoplankton blooms and decaying macroalgae on the shores of the island, the inhabitants of Utö are known to
suffer from a very bad smell. In our measurements, the odour threshold was exceeded often in the summer of 2018.
Novak et al. (2022) found that methanethiol emissions are dependent on wind speed, but in this study, we did not find any
correlation of mixing ratios with wind speed. While wind might increase emissions, it also increases dilution in the air. In
addition, there are several other factors that impact the mixing ratios, e.g., oxidation and mixing layer height, and therefore
emissions and the factors impacting them are not expected to be directly comparable with mixing ratios.
In the atmosphere methanethiol oxidizes with OH radicals seven times faster than DMS (Kilgour et al. 2022). OH oxidation
of methanethiol produces $SO_2$ with almost unity yield (Novak et al. 2022). It has been estimated that methanethiol could be a
source of up to 30% of the $SO_2$ formed in the marine boundary layer in coastal California, where clearly lower mixing ratios



were measured than in this study (Novanak et al. 2022). In our study the methanethiol mixing ratios were high over several months and not just during short blooming periods (Fig. 2). This indicates that methanethiol could have a stronger contribution

on $SO_2$ production in this area. More studies on methanethiol emissions would be needed to confirm this.

## 4 Conclusions

The ambient air mixing ratios of NMHCs and methanethiol were studied at the marine research station on Utö island in the

Baltic Sea. The NMHC mixing ratios were typical for the northern rural/remote site. The seasonal variations of NMHCs followed a well-known cycle with maximum mixing ratios in winter and minimum during summer. The exception was methanethiol, which was identified here for the first time. It had a clear maximum in spring and summer.

Especially for longer living NMHCs, the regional background was shown to be the major source. The contribution of the background had an exponential correlation with the lifetime of the measured alkanes and aromatic hydrocarbons. This gives

confidence that PMF can produce valid information on the source apportionment of NMHCs also at rural/remote sites, where the ratios of these reactive compounds have been altered during transport from their sources to the site.

Of the local/regional sources, shipping had a strong impact especially on ethene, propene, ethyne and benzene. For shorter living NMHCs (aromatic hydrocarbons and 2-methylpentane), traffic emissions had major effect. Wind distribution analyses indicated that these traffic emissions came from the direction of the main harbours/cities of the Gulf of Finland including the

city of Saint Petersburg 500 km to the east of site. Gasoline evaporative emissions originating in the east had a strong impact on butane and pentane levels.

The mixing ratios of methanethiol followed the variations of seawater temperatures from the end of April until mid-July. After that the mixing ratios started to decline while seawater temperatures remained high. The diurnal variation of the mixing ratios still followed the variation of the temperatures. During that time the abundance of phytoplankton in the seawater also declined.

The mixing ratios also negatively correlated with sea level height especially in May and August. Macroalgae exposed to ambient air during low sea levels may start decaying faster and induce methanethiol emissions. This together with ambient air temperature dependence and high summertime mixing ratios indicated the biogenic origin of methanethiol possible resulting from phytoplankton or macroalgae. The detected mixing ratios were higher than found earlier in other areas and may be the source of the malodour detected on the island during strong phytoplankton blooms. This may also have strong impacts on local

$SO_2$ production and new particle formation. More studies on methanethiol emissions and atmospheric impacts would be needed.

*Competing interests:* The contact author has declared that none of the authors has any competing interests.

*Acknowledgements:* The observations at Utö were supported by Academy of Finland project SEASINK (grants # 317297& 317298), Finnish marine research infrastructure (FINMARI) and the JERICO-S3 project, funded by the European



Commission's H2020 Framework Programme under grant agreement No. 871153. JPJ would like to acknowledge the funding from the EU H2020 project EMERGE, which received funding from the European Union's Horizon 2020 Research and Innovation Programme under grant agreement no. 874990 (EMERGE project). Timo Mäkelä and Juha Hatakka are thanked for their help on setting up the measurements. Milla Johansson is thanked for providing sea level height data. Simo-Matti Siiriä is acknowledged for drawing the Fig. 1.

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

625   .