# Peer review of "Shipping and algae emissions have a major impact on ambient air mixing ratios of NMHCs and methanethiol on Utö island in the Baltic Sea"

_EGUsphere, 2023_

## Author Response (AR1)

Dear editor,

We have carefully considered all the reviewer comments on our manuscript 'Shipping and algae emissions have a major impact on ambient air mixing ratios of NMHCs and methanethiol on Utö island in the Baltic Sea' and they have helped us improve the manuscript. We provide here below answers and mention the changes that have been made to address the referee's concerns. The referee's comments are in normal font while the replies are given in red.

Yours sincerely,

Heidi Hellén

**Answers to the reviewer 1 comments:**

The author conducted long-term observations of VOCs on UTO Island, attempting to illustrate the effects of shipping and algae on VOC concentrations. The author's article is based on two themes, namely VOCs and methyl mercaptan. The changes and sources of these two substances have extremely weak correlation, which leads to the article being disorganized. This also reflects that the author wants to study each topic, but the research on each topic is not in-depth. The details of the research are unclear and the purpose of the research is confusing, so I suggest rejecting the manuscript.

- we hope that new version of the introduction describes the aims of the study better. Indeed, anthropogenic VOCs and biogenic ones such as methylmercaptan, have very different sources and variability, but still both of them have an important contribution to marine atmosphere.

Major comments

I suggest the author conduct in-depth analysis on one topic, rather than discussing both topics in a general manner.

1. The author's research is mostly based on methods rather than purposes. For example, the author mainly wants to study the sources of VOCs on UTO Island and determine that the main source is shipping. Why do you conduct CBPF and SILAM analysis? These two types of research do not seem to affect the main conclusion of the article.

   - CBPF and SILAM were used for the interpretation of the PMF source factors. Interpreting PMF results of these reactive compounds at this background site is not straight forward, since compounds ratios may change during the transport. CBPF and SILAM gave us additional information on the possible source areas and distance of the source from the site, which helped us on the interpretation. Figure caption of the figure 4 was improved to better describe the information given by the CBPF figures.

2. Because the author studied two topics, the introduction section was written separately, which resulted in a confusing logic in the introduction section. The author should write in sections according to the research significance, research progress, existing problems, and the work to be carried out.

- Introduction was reconstructed and improved.

3. Methods section. How many substances have the author detected in total? At least there needs to be a list.

   - List of measured compounds was added to the section 2.2

4. Section 3.1 The seasonal changes of each substance by the author are not the main content. What is the minimum proportion of each substance that the author needs to present? Which substances are the top 10? The sorting of these substances is highly helpful for subsequent source apportionment judgments.

   - We renamed section 3.1 as 'Mixing ratios of NMHCs' and more discussion on the results were added into this section. However, since many studies on mixing ratios of NMHCs has been published earlier, we were not focusing more on this. We also hope that new version of introduction presents the aim of this study now better.

5. In section 3.2, how many substances were selected by the author for PMF work? How much do these substances account for in total VOCs concentration? Background sources are generally characterized by chlorinated hydrocarbons. Have these substances been detected in this study?

   - We added to the manuscript section 2.3 'All compounds quantified, except p/m-xylene and methanethiol, were used for the PMF analyses. Methanethiol remained below detection limit during the winter months and p/m-xylene comprised < 2% of the total detected VOC mixing ratio.' Unfortunately, chlorinated compounds were not quantified in this study.

6. Section 3.2. Why is there no biomass combustion source? Natural gas combustion source? Industrial source? Plant source? Europe burns a lot of natural gas, at least there should be a contribution from natural gas combustion.

   - This is a remote site, where these sources may have been diluted and mixed into regional/global background.

7. The results of source parsing need to be well validated. For example, can you verify it through the time series shown in Figure 6. Is the contribution of different source classes in Figure 6 related to the level of activity? For example, why is background contribution more significant in November? Why is the contribution of transportation more significant in February? Why is local contribution more significant in January? Can the author provide relevant evidence? In addition, diurnal variation may also characterize the contribution of different source types. Generally, sources with strong convective activity during the afternoon will have more regional contributions. The author can also use this method to demonstrate the accuracy of source analysis results.

   - Due to very low local activities, contribution of the factors are expected to vary based on the wind directions and routes of the air masses, which are shown by the wind direction distributions (Fig. 4) and source area estimates (Fig. 5). Diurnal variation of the sources was calculated. Diurnal variation was not strong. This was expected due to low local sources and activities. Strongest variation was observed for the F5 (Local solvent) (see figure below). For it highest contributions were measured during the night when the mixing layer heights are generally low and

therefore local emissions are not diluted as much and may be accumulated. The lowest (35 % lower than the highest value at 4:00) contributions of F5 were in the afternoon at 18:00. F1 (Gasoline) and F2 (traffic exhaust) had ~20% higher contributions during the mid-day (10:00-14:00) compared to other times of the day. This could represent local daytime activity. F3(background) had lowest contributions during the night. Since diurnal variation was so low and results agreed with our other factor interpretations, we did not add this into manuscript.

[Figure]

Figure: Mean diurnal variations of factor contributions normalized by the maximum contribution

8. The author's research on methanethiol has significant shortcomings. The author is unable to explain the relationship between seawater temperature and methylthiol in August and September very well. The author used planktonic biomass to explain, which made me feel that planktonic biomass is the main factor leading the emission of methanethiol. However, the author devotes a significant amount of space to describing the relationship between seawater temperature and methanethiol. I think the author needs to draw a cautious conclusion and delve into the dominant factors behind the changes in methanethiol concentration.

- we are aware that complementary measurements for methanethiol are incomplete, but we still hope to publish these results so that more research could be focused on this, and that other global marine stations measuring with the similar set-up (e.g. stations within the Global Atmospheric Watch Program), could start looking for this compound, which may have significant emissions and atmospheric impacts, even with very low mixing ratios detected.

- To improve the description of the relations of methanethiol with sea water temp and total phytoplankton mass we modified the figure 8. Even if phytoplankton is the source of the methanethiol, temperature may be significant factor controlling its production by phytoplankton and its transfer from sea to atmosphere and is therefore important. We modified the text in the section 3.3 to be clearer that we hypothesize that especially during summer methanethiol is coming from phytoplankton, but more research would be needed to confirm this. We removed the figure 9 and discussion related to that since it was highlighting too much the role of temperature.

**Answers to the reviewer 2 comments**:

In the manuscript „Shipping and algae emissions have a major impact on ambient air mixing ratios of NMHCs and methanethiol on Utö island in the Baltic Sea", Hellén et al. measured atmospheric NMHCs as well as methanethiol for one year from March 2018 to February 2019. They report about seasonal changes of NMHCs as well as identified drivers (compounds) for 5 different factors (Gasoline fuel, traffic exhaust, background, ship emissions, local solvent) using PMF. Furthermore, they calculated source area estimates using atmospheric dispersion simulations.

This study about NMHC concentrations in the Baltic Sea is an important work to better understand the composition of different atmospheric substances with respect to their origin. Furthermore, it is important to understand source regions especially from anthropogenic input. The study itself is well laid out and the presentation of the data is mainly satisfactorily but needs some more information here and there. I suggest to publish this important work after addressing the following comments.

Major comments:

The introduction seems randomly thrown together. There are very short paragraphs (a few lines each) about very, very basic information about NMHCs, followed by information about previous measurements of NMHCs at the study site, followed by information about PMF applications mixed together with information about the current study, followed by a random sentence about other pollutants/combustion by ship engines, followed by information about the Baltic Sea with respect to biological measurements as well as related information about DMS and methanethiol. Please rewrite and restructure the introduction with respect to the main messages of the manuscript. The introduction should give a state-of-the-art current knowledge of the topic which is investigated. Information about the work which has been done within this study should be mentioned at the end of this section and not bit wise within all the different paragraphs.

- Introduction was reconstructed to better describe the current knowledge and aims of the study.

Seasonal variations of NMHCs: I am wondering if the authors did miss to provide any text (apart from 2 sentences) about the seasonal variations of the 15 different compounds? This section is called results and discussion, but I do not see any written results or discussion about seasonal variations.

- We renamed section 3.1 as 'Mixing ratios of NMHCs' and more discussion on the results was added into this section. However, since studies on mixing ratios of NMHCs have been conducted at several locations with many publications, we were not focusing more on the mixing ratios. We also hope that new version of now introduction presents the aims of this study now better.

I am wondering if the manuscript would be more coherent if methanethiol (MeSH) results would not be part of this manuscript. It seems a bit random that MeSH is included (as the authors say: this peak was found during the campaign and it was not planned to measure it). MeSH is biogenically produced in the ocean and therefore totally different in comparison to the other presented NMHCs. Furthermore, biogenic data which could partly explain seasonal changes of MeSH are only available for 2 out of 12 months. Here I would like to cite lines 353-355 in the manuscript: "The reason for these differences could be shifts in phytoplankton community composition and their physiological

status, possible contributions from macroalgae vegetation from shoreline, and meteorology." This shows pretty clearly that the authors are aware that no complementary measurements have been performed during this study which could be used to explain the measured MeSH concentrations. Therefore, I would suggest to totally disentangle these results from the main objective of this manuscript.

- In our opinion results on this currently unaccounted compound are highly interesting and are expected to have strong impacts on new particle and cloud formation in the atmosphere. It is possible that this compound has been missed globally also at other marine station. Measurements of this compound were not in our original project plan and therefore we did not conduct proper complementary measurements, but publishing these preliminary results, could encourage other researchers and station using similar methods for NMHC monitoring start looking at this compound. One reason why global NMHC community may have generally missed this compound is its high reactivity. Due to this high reactivity, even with relatively high emissions, mixing ratios are expected to remain low. We now tried to state the importance of these results better in the introduction. Even though we were not able to produce more complementary data for this, we now tried to improve presentation of existing data by modifying the figure 8.

Specific comments:

ll.30: The first paragraph has only one reference. Please be more specific with the references related to the statements within the text.

- we added more references to the introduction

ll.176: It would be nice to read how these complementary meteorological data has been measured.

- we added a short description on the methods for measuring meteorological data, ozone ($O_3$), sulphur dioxide ($SO_2$), particulate matter <2.5 µm ($PM_{2.5}$) and nitrogen dioxide ($NO_2$) and reference to the figure 1 showing the location of the measurements.

Figure 2: Do the authors check for outliers within each dataset (month)? Whiskers of different compounds and month show highest and lowest measurement, however, it would be interesting to know from a statistical point of view, if these values are outliers or not. Furthermore, it would be interesting to know from how many datapoints each boxplot is generated, especially when there are only limited values above LOD in winter time.

- Outliers were not removed from the original data set for figure 2. Using the statistical method 3*IQR, NMHC ratios and visual inspection, we removed extreme outliers and plotted new figure 2. Number of data points was 2175- 2188 for each compound. For values <LOD we used the value of 0.5*LOD. This information was added into figure caption 2.

Figure 3: Please be consistent with naming different factors F1-F5 throughout the manuscript. Here, e.g. F2 is mentioned "Local traffic" and "local exhaust" in Fig 3a and Fig 3b. In Table 1 F2 is listed as "traffic exhaust". This is very confusing when reading through the manuscript. I suggest not to use "local" as "local" is already part of F5.

- As suggested by the reviewer we kept "traffic exhaust" as a naming of F2. Naming of F2 was corrected into the figure 3.

Figure 4: What kind of information do the circles provide? It is very hard to read but they show values like 5, 10, 15 and 20. What do they stand for?

- explanation that circles show the wind speed with 5 m/s increments was added to the figure caption.

Figure 6: Wouldn't it make sense to include the wind direction in this Fig as wind direction plays also a critical role in source area probability?

- wind direction was added into figure 6.

ll.315: If methanethiol will be included in the manuscript it needs a proper part in the introduction and not in this paragraph.

- Introduction on methanethiol has been improved and moved to the introduction section.

ll.335: Error bars for every time step in Fig 9 are missing to understand the significance of the variability. Are the correlations significant? Please provide some statistics for the diurnal analysis. Additionally, it could help to normalize the diurnal data to account for strong changes in absolute concentration within a month when averaging each hour of a month.

- since figure 9 was highlighting too much the role of the temperature, we removed the figure based on the comments by the reviewer 1.

Figure 10: Both blueish colors for June and September are hard to distinguish (background color of box for equations)

- we changed the colouring of the figure 10